# Combined LTP Sublingual and Oral Immunotherapy in LTP Syndrome: Efficacy and Safety

**DOI:** 10.3390/jcm12051823

**Published:** 2023-02-24

**Authors:** Maria Aranzazu Martin Iglesias, Rosa Garcia Rodriguez, Alberto Palacios Cañas, Jaime Vinicio Meneses Sotomayor, Miriam Clar Castello, Francisco Feo Brito

**Affiliations:** Allergy Department, General University Hospital of Ciudad Real, 13005 Ciudad Real, Spain

**Keywords:** LTP lipid transfer protein, sublingual immunotherapy, oral immunotherapy, LTP syndrome, LTP desensitization, quality of life, FAQLQ-AF

## Abstract

Introduction: SLIT for the treatment of plant food allergies has been demonstrated to be safe but less effective than OIT, but the latter is associated with more adverse reactions. The aim of the study was to evaluate the efficacy and safety of a new protocol starting with SLIT-peach followed by OIT with commercial peach juice in patients with LTP syndrome. Methods: This was a prospective, noncontrolled, open study on patients with LTP syndrome who are not sensitized to storage proteins. SLIT peach ALK was followed by OIT with Granini^®^ peach juice after 40 days of the SLIT maintenance phase. At home, the Granini^®^ juice dose was progressively increased during the 42 days until reaching 200 ml. After achieving the maximum dose, an open oral food challenge was carried out with the food that had caused the most severe reaction. If negative, the patient was instructed to progressively introduce the foods that were avoided before starting immunotherapy at home. Patients were reviewed 1 month later. The quality-of-life questionnaire FAQLQ-AF was completed at the beginning of the study and one month after the final challenge. Results: Forty-five patients were included, most of them with LTP anaphylaxis. Peach SLIT was well tolerated in 80.5%, and OIT with Granini^®^ was well tolerated in 85%, with no severe adverse reactions. The final provocation was successful in 39/45 (86.6%). One month after the final provocation, 42/45 (93.3%) patients had no dietary restrictions. FAQLA-AF was significantly reduced. Conclusions: This combination of peach SLIT and OIT with commercial peach juice provides a new, fast, effective, and safe immunotherapy option for selected patients with LTP syndrome who are not allergic to storage proteins, improving their quality of life. This study suggests that cross-desensitization relative to the nsLTPs of several plant foods can be achieved by using Prup3.

## 1. Introduction

Nonspecific lipid transfer proteins (nsLTPs) are the main cause of primary food allergy in adults living in Mediterranean European countries [1]. nsLTPs belong to the superfamily of prolamins, which are widely distributed throughout the plant kingdom and are classified as PR-14 proteins [2].

LTP is a major allergen in the *Rosaceae* family, and nsLTPS are relevant allergens in several pollens and in many botanically unrelated foods such as fruits, vegetables, nuts, legumes, cereals, spices, and seeds. 

Patients with LTP syndrome suffer multiple plant food allergies (mainly toward fruits and nuts) caused by sensitization to different plant foods nsLTPs, because of the cross-reactivity between them, as they share linear and conformational epitopes [3]. 

The clinical expression is extremely variable, ranging from patients who suffer anaphylaxis to others that react only in the presence of cofactors such as exercise [4], NSAIDs, or alcoholic beverages. Allergy toward nsLTPs may provoke in the same patient symptoms relative to many plant foods, including those that are raw, cooked or processed [5], because nsLTPs are resistant to heat treatment and low pH [6].

Thus, this syndrome may cause stress and limitations in life patterns for patients and their families due to the need to be alert with their diet and the fear of suffering reactions with the inadvertent ingestion of foods containing LTP, which can give rise to a substantial impairment on their quality of life, that can be measured by the validated Food Allergy Quality of Life Questionnaire (FAQLQ) [7].

Within the natural history of LTP syndrome, an increase in the number of reactions to novel foods has been observed [8,9,10]. This phenomenon cannot be predicted by nsLTP-specific IgE [11].

In recent decades, studies have been carried out using sublingual (SLIT) and oral (OIT) immunotherapy with foods as an alternative to restrictive diets, producing encouraging results. 

The mechanisms by which immunotherapy induces tolerance are unclear. OIT may alter the T cell responses by the deletion of pro-allergic T cell responses, by the immune deviation from a Th2 to Th1, or by the induction of concurrent immune-regulating T cells. Dendritic cells may also play a role in OIT outcomes [8]. 

On the other hand, current data show that antigens delivered by SLIT are taken up by a myeloid dendritic cell population in the oral mucosa, oral Langerhans cells. This promotes the T-cell production of tolerogenic cytokines, including IL-10 and TGF-β. FOXP3-positive T regulatory cells are induced, and Th2 cytokines, including IL-4, are downregulated, increasing Th2 cytokines (IFN-γ and IL-12) [9,10].

In 2005, Enrique et al. [12] published the first clinical trial of SLIT with a vegetable food: hazelnut. They obtained a significant increase in the dose of hazelnuts tolerated compared to the placebo group, with very few systemic reactions. Similar results have been published with other foods, such as peanuts [13]. 

In 2009 Fernández–Rivas conducted a clinical trial of SLIT with a peach extract quantified in mass units for Pru p 3 [14], demonstrating an increase in the mean tolerance threshold for peaches after 6 months of treatment, with all reactions being mild. 

Since 2011, sublingual immunotherapy with the allergen Pru p 3 has been commercially available. This immunotherapy (SLIT-peach ALK^®^) has proven to be an effective treatment not only against peach allergy but also against allergies to tree nuts and peanuts. Gomez et al. [15] demonstrated the possibility of cross-desensitization between the LTP of different plant foods.

Beitia et al. [16], in 2021, also obtained promising results with Pru p3 SLIT, while 53% of the control group experienced reactions to new foods throughout the follow-up period, and the severity of symptoms increased significantly. 

The use of SLIT followed by OIT could reduce reactions in the induction phase by starting from a greater tolerance at the beginning of OIT; in addition, it could allow for achieving a higher level of tolerance to the food or even a free diet. 

Most protocols of oral immunotherapy use natural foods with proven efficacy [17]. Based on these protocols, we can use fresh peaches as well, but given the difficulty of obtaining fresh peaches throughout the year, we planned to use commercial peach juice (Granini^®)^ as an affordable and cheap source of LTP. 

Navarro et al. in 2019 [18] assessed the safety and efficacy of oral immunotherapy with Granini^®)^ peach juice (21.16 µg/mL LTP concentration) in an initial pilot study on 19 patients; 17 achieved tolerance relative to 200 mL after 18 visits to the immunotherapy unit after a mean time of 3.6 months. 

Desensitization is quite specific; Gomez [15] suggested the possibility of achieving tolerance to other plant food groups using peach LTP as the source for OIT, but, to date, this hypothesis has not been proven. 

The aim of the study was to evaluate the efficacy and safety of a new protocol starting with SLIT-peach and followed by OIT with commercial peach juice in patients with LTP syndrome and to assess changes in the quality of life throughout the study.

## 2. Methods 

This was a prospective, noncontrolled, open study carried out over a year in the Allergy Section of the Hospital General of Ciudad Real, Spain. 

The study was approved by the Center’s Ethical Committee (C-463).

### 2.1. Study Population

Patients 16 years and older diagnosed with LTP syndrome and who gave their informed consent were recruited for the study.

Inclusion criteria included a documented history of moderate or severe reactions occurring less than one hour upon eating two or more different plant foods within the last 6 months, with or without the intervention of cofactors, and that was caused by allergy to LTP. This was demonstrated by positive skin prick tests (SPT) and/or specific IgE (IgE ≥0.35 kU/L ImmunoCAP FEIA Thermo-Fisher, >0.3 kU/L ISAC or ALEX) relative to LTP in the patient’s sera.

We excluded pregnant women, patients with another severe disease, poorly controlled asthma, or atopic dermatitis. Patients whose reactions were also mediated by other relevant vegetable panallergens, such as storage proteins, were also excluded. Patients sensitized to profilin were not excluded as it rarely triggers systemic reactions.

### 2.2. Study Protocol

All participants underwent (Figure 1) a baseline evaluation (visit 0) that included a complete medical history, with a detailed description of the reaction and graded anaphylaxis according to the WAO [19]. SPTs were performed with all the foods involved in the reactions according to the clinical history and with the panallergens profilin and LTP (Leti laboratories). Prick-by-prick tests with fresh vegetable foods were also performed when available. SPTs were performed according to the EAACI recommendations [20].

The patient’s sera were analyzed for the total IgE and for multiplex microarray techniques (ISAC or ALEX) in order to obtain their sensitization profile at the molecular level.

Informed consent for the described protocol was obtained for each subject as well as a completed FAQLQ-AF quality-of-life test.

### 2.3. Sublingual Immunotherapy

The concentration of SLIT peach ALK^®^ is 50 µg/mL. The induction phase was carried out with a two-day adjusted outpatient regimen in the Allergy Day Hospital (visit 1) (Figure 2). 

In case of a reaction, the SLIT induction phase was extended to 3–4 days.

After the induction phase, they continued at home with a daily sublingual 4 drops regimen (10 µg) for 40 days. 

### 2.4. Oral Immunotherapy with Granini^®^ Peach Juice 

After finishing the programmed SLIT peach, the patient came back to the Allergy Day Hospital (visit 2). The number and severity of the reactions during SLIT peach ALK were recorded.

OIT started with increasing doses of Granini^®^ peach juice at a concentration of 21.16 µg/mL [18], which was demonstrated to have a little variation among different bottles and batches. [18]. Doses were increased until reaching a dose of 4.1 mL in 1 day at the hospital (Figure 3). SPT with LTP and a prick-by-prick test with Granini peach juice were performed. 

Once the dose of 4.1 mL had been reached, the patient was instructed to take Granini^®^ at home with daily 10% incremental doses, throughout 42 days, until achieving a dose of 200 mL, which equals a medium piece of peach, since 1 peach weighs about 100 g and Granini^®^ juice has a 50% fruit content (information provided by Eckes–Granini Ibérica).

Both sublingual immunotherapy and Granini^®^ were purchased by the patient.

### 2.5. Final Provocation Test (FPT)

Once the daily dose of 200 mL of Granini^®^ peach juice had been reached, an open oral challenge test was carried out at the Allergy Day Hospital (visit 3) on the following day with the food that caused the most severe reaction. When several foods triggered a similar reaction, we decided on one to carry out the test, in agreement with the patient and according to his/her preferences. The food was given in three doses [21] (Figure 4).

The FPT was considered positive if an objective sign or symptom was detected (urticaria, angioedema, vomiting, bronchospasm, desaturation, or hypotension) or intense and prolonged (more than 30 min) abdominal pain occurred during the observation period. In our series, all patients who reacted in relation to cofactors also had symptoms without the presence of cofactors. However, in these cases, the provocation was considered negative only when the patient ingested the food without any restriction (exercise and NSAISs) throughout the following month after a negative hospital provocation test. 

In the event of a negative FPT, the patient was recommended to gradually introduce the foods avoided before immunotherapy at home and to continue taking Granini^®^ peach juice at least 3 days a week for 6 months. Cofactors were not forbidden.

In the case of a positive FPT, the patient continued taking Granini^®^ juice daily until a new challenge was carried out one month later with another food that also caused a severe reaction. The implicated food was rechallenged in the following 6 months.

### 2.6. Follow-Up

Patients were reviewed 1 (visit 4) and 5 months (visit 5) after the final challenge. During these visits, clinical evolution was assessed by recording the tolerance of the introduced foods. SPTs with LTP and Granini^®^ were repeated on visit 4.

FAQLQ-AF was filled in again during visit 4.

### 2.7. Statistical Analysis 

All analyses were performed using the SPSS package (PASW Statistics 18). Quantitative variables were described as the mean and standard deviation, whereas categorical ones were presented as frequencies and percentages. The t-test was used to compare different groups, and a 1-sample paired t-test was used to compare quantitative variables at different study time points if they followed a normal distribution. If this was not the case, non-parametric tests, such as the Wilcoxon and Mann–Whitney U-tests, were used. 

## 3. Results 

Fifty patients were recruited; forty-five patients completed the study. One patient did not start immunotherapy because of pregnancy, another did not start due to a transfer of residence, and the others abandoned peach SLIT at the beginning of the protocol by their own decision due to irregular adherence.

### 3.1. Baseline Characteristics of Treated Patients 

Of the 45 patients treated, 27 were women, with a mean age of 32.29 years (range 16-63); (see baseline characteristics in Table 1). Molecular diagnosis was performed for 42 patients, with 28 patients using ALEX and 14 patients using ISAC.

All patients presented symptoms with multiple foods: 66.7% with ≥3 groups of foods. The foods that caused the most severe reactions were nuts (31.1%) and peach/*Rosaceae* (almond included) (26.7%). All patients had reactions without the intervention of cofactors, but in 10/45 patients (24.4%), reactions were sometimes triggered sometimes in the presence of cofactors, with exercise being the most frequent, followed by NSAIDs.

Eighty-four percent of all patients reported symptoms of oral allergy syndrome (OAS); another 88.9% presented skin symptoms. Thirty-four patients (75.6%) suffered anaphylaxis; the most frequent grade was grade IV (44.1%).

The mean FAQLQ-AF score was 4.47.

### 3.2. SLIT and OIT 

The induction phase with peach SLIT was completed following the manufacturer’s instructions. In 41/45 treated patients, we used a 2-day induction phase, extending it to 3 days in one patient and to 4 days in three patients due to mild reactions. 

Peach SLIT was well tolerated in 82.2% of the patients, with five patients presenting OAS during induction and three presenting OAS and generalized pruritus: one patient during induction and two patients during the maintenance phase (Table 2).

The transfer to Granini^®^ was also carried out at the Hospital, followed by daily increasing doses at home. A dose of 200 mL was reached in 42 days, with good tolerance in 86.6%; six patients presented OAS, and one was accompanied by generalized itching. The dose of 200 mL could be achieved in all patients.

### 3.3. Final Provocations and Follow-Up 

The FPT was successful in 39 patients (86.6%) (Table 2). Of the remaining six patients, three presented a mild reaction, two had a moderate reaction, and one presented a severe one.

The three mild reactions occurred with almond (*Rosaceae*), apple-with-peel (*Rosaceae*), and peanut (legume), and the reaction comprised oral itching and isolated wheals that resolved with antihistamines and oral corticosteroids. In the first two patients, we repeated the FPT one month later with good tolerance and the third patient tried it at home with no reaction.

The two moderate reactions occurred with almonds and apple-with-peel 1–2 h after the total dose and consisted of generalized, intensely itchy urticaria requiring treatment with antihistamines, parenteral corticosteroids and finally, epinephrine due to scarce improvement. In these patients, a provocation test with another *Rosaceae* was carried out (pear-with-peel and almond, respectively) with good tolerance. They currently follow a free diet, except for the food involved, which will be proven within 6 months from the FPT.

The severe reaction was caused after the second dose of lentils, presenting generalized urticaria, angioedema and digestive symptoms that resolved with antihistamines, corticosteroids, and adrenaline. One month later, a provocation test with apple-with-peel exhibited positive results with mild urticaria 20 min after eating the entire piece. The patient continued eating small amounts of any food with no reactions.

In the 1-month visit after the FPT, 42/45 had no dietary restrictions, 2 patients followed a free diet except for the food involved (almonds and apples) and the patient with the severe reaction continued with dietary restrictions, eating only small amounts of the foods involved.

Six months after the FPT, three free-diet patients presented mild reactions at home to nuts, one of them coinciding with several cofactors. Two of them rejected new provocations because, apart from those foods, they followed a free diet. The third one tolerated it at the hospital one month later and followed a free diet.

Among patients with a negative provocation, two patients worked in a fruit store; one of them exhibited symptoms after ingestion of and contact with fruits and, after immunotherapy, followed a free diet but maintained contact symptoms, although they are much milder. This patient, after immunotherapy, only had domestic contact with fruits because she no longer works in the fruit store. The second patient presented symptoms of asthma and urticaria and, after immunotherapy, continued to carry out her work with excellent tolerance. 

Another patient is a cook who also improved his contact symptom and follows a free diet. 

There were no significant differences in the SPT mean wheal with LTP or Granini^®^ juice between baseline and visit four when using the Wilcoxon test. There were either no differences in SPT with LTP or with Granini^®^ or in sIgE with LTP between those who successfully passed the FPT and those who did not (Mann–Whitney U-test). 

One month after de FPT, following a free diet (42/45) or with the limitations recommended after the challenge, the FAQLQ-AF score was significantly reduced from the initial 4.47 points to 2.7 points (Wilcoxon test) (Table 3).

## 4. Discussion

Our study shows that patients with the LTP syndrome can achieve tolerance relative to various groups of plant foods with the described protocol combining SLIT peach and ITO with peach juice, allowing a free diet in a very high proportion of patients and with very few adverse effects.

After this rapid and well-tolerated protocol, a significant improvement in the quality of life of all patients was detected, as measured by FAQLQ-AF.

We observe that cross-desensitization between plant foods is possible using peach LTP as a natural source for OIT, as 70% of the patients were provoked with group foods different from the *Rosaceae* family. Interestingly, of the 6/45 who reacted in the FPT, two patients reacted to almonds, and two patients reacted to apple-with-peel, both of the *Rosaceae* family.

It would seem from these data that cross-desensitization is at least as effective for plant foods from groups other than *Rosaceae*, which probably depends on the linear or conformational epitopes that are specifically shared between each food LTP and peach LTP.

As previously mentioned, Gomez [15], Navarro [18] and Beitia [16] achieved cross-desensitization with nuts after immunotherapy with Pru p3. We demonstrate that it could also occur with other groups of vegetable foods.

All patients who tolerated the food in the FPT were able to introduce all those that previously caused them reactions. 

The cofactor-related reactions in our series are lower than those observed in other published articles [22,23,24], and all our patients had reactions without the presence of cofactors; thus, so if the provocation had negative results, we allowed the patients a free diet without avoiding cofactors.

Regarding the two patients working in the fruit store, we hypothesized that the better evolution of one of them might be due to the daily and more intense exposure to fruit that would favor the maintenance of tolerance. The cook also improved his contact symptoms, besides being able to follow a free diet. 

These last cases suggest that this immunotherapy is efficacious not only in patients with reactions after the ingestion of foods but also in those who present symptoms upon inhalation or contact. 

In disagreement with other studies [14,15], the SPT with Pru p3 and with Granini^®^ commercial juice did not offer significant differences from the beginning of the study to one month after the FPT. 

Similarly to other studies, specific IgE levels do not predict severity [11]. In fact, neither the SPT with LTP or with Granini^®^ nor the sIgE with LTP exhibited higher figures in those who reacted with the FPT, which suggests the inadequacy of these variables when predicting the patient’s response to the treatment, as described in other studies [25].

The included patients had a highly impaired quality of life because of dietary restrictions, and the fear of a serious reaction since 34 patients (75.5%) had presented anaphylaxis, most of them grade III-IV. In other studies, with less severe patients [12], the obtained results were worse in terms of efficacy. 

One of the limitations of our study is the lack of controls, but according to several authors [11,16,26], the natural evolution of the LTP syndrome progressively worsens, reacting to more vegetable foods; thus, tolerance achieved after the combination of SLIT and OIT with Pru p 3 could not be justified by a natural evolution of this syndrome, and symptoms are not expected to resolve spontaneously in such a short period of time. The published studies may be considered an alternative to the controls. Patients were not challenged before treatment for ethical reasons, as a high proportion manifested anaphylaxis; the history was unequivocal, and the patients reported reactions in the previous 6 months. 

The proposed protocol has the advantage of being quicker than others, and it can be completed in 82 days, with fewer hospital visits.

Our study includes a greater number of patients than those previously published in studies with LTP syndrome [12,13,14,15,16,18]; furthermore, they were selected with well-defined selection criteria after performing a molecular diagnostic study. We excluded patients sensitized to storage proteins; this was required because storage proteins can be found in the implicated plant foods and can cause severe reactions, rarely occurring with profilin. This exclusion criterion may have contributed to the success of our protocol since patients allergic to nuts or legumes could also be allergic to storage proteins for which the patients were not treated. 

Another important advantage of our study is that quality of life was measured using a validated FAQLQ-AF before and after the combined immunotherapy.

We conclude that our study provides a new, quick, effective, and safe immunotherapy option for patients with LTP syndrome without sensitization to storage proteins, and an improvement in their quality of life has been demonstrated with the described protocol.

This study suggests that cross-desensitization with OIT between nsLTPs from different families of plant foods is possible, as tolerance to LTPs from nuts, legumes, and other non-*Rosaceae* fruits has been achieved using Pru p3, the peach LTP.

Given the safety and little time consumption of the protocol, we consider it a highly recommended option in well-selected patients with reactions due exclusively to LTP.

## Figures and Tables

**Figure 1 jcm-12-01823-f001:**
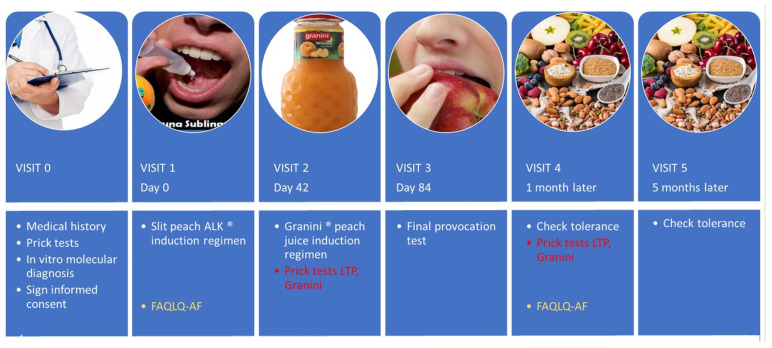
Timeline, study chronology. FAQLQ-AF: Food allergy quality of life questionnaire-adult form.

**Figure 2 jcm-12-01823-f002:**
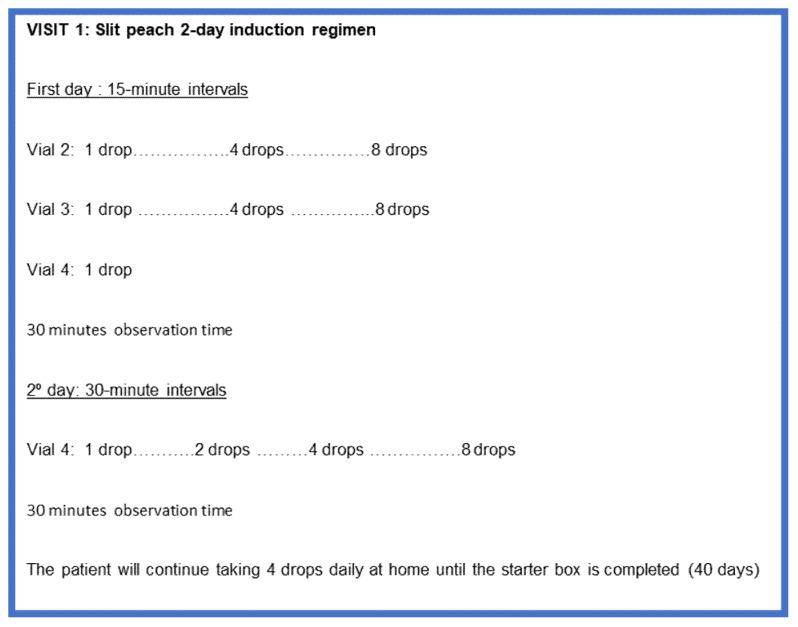
Visit 1: SLIT 2-day induction phase.

**Figure 3 jcm-12-01823-f003:**
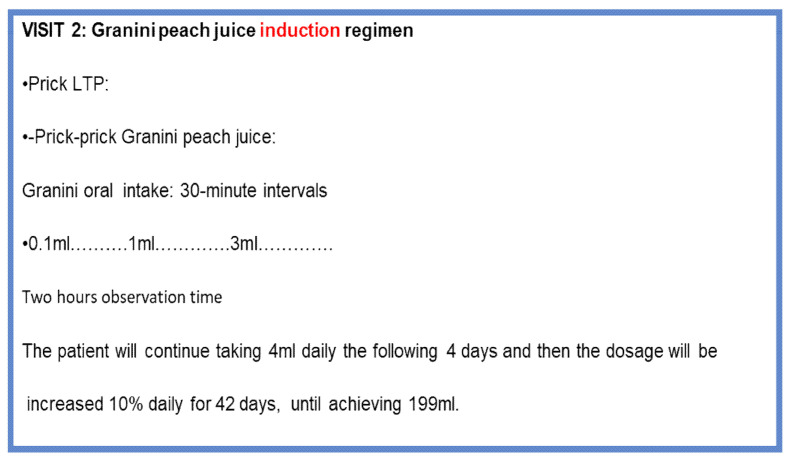
Visit 2: OIT induction regimen.

**Figure 4 jcm-12-01823-f004:**
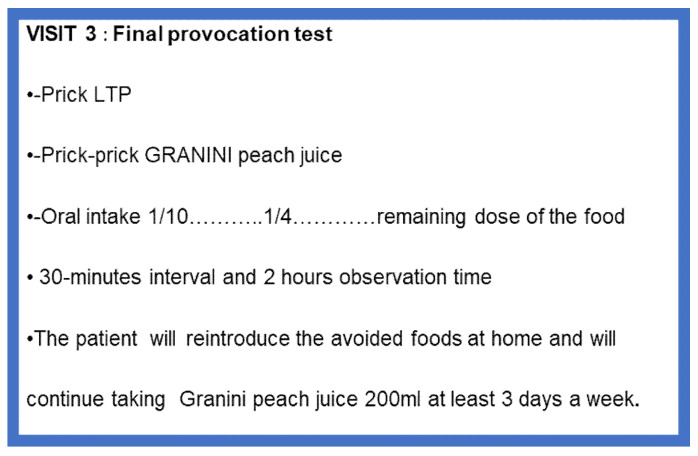
Visit 3: Final provocation test.

**Table 1 jcm-12-01823-t001:** Baseline characteristics of the patients.

N	45
Age, years (mean (SD))	32.29 (12.6)
Gender (F/M)	27/18
Family history of atopy n (%)	38 (84.4%)
Personal history of atopy n (%)	32 (71.1%)
Associated pollen allergy n (%)	34 (75.6%)
Rhinoconjuntivitis n (%)	35 (77.8%)
Asthma n (%)	22 (48.9%)
Food groups involved n (%)	
Several Nuts	1 (2.2%)
Several Fruits	2 (4.4%)
Two groups of plant foods	12 (26.7%)
Three groups of plant foods	14 (31.1%)
≥three groups of plant foods	16 (35.6%)
Food that caused the most severe reaction n (%)	
Nuts	14 (31.1%)
Peach/Rosaceae	12 (26.7%)
Legumes	4 (8.9%)
Another fruits	4 (8.9%)
Grains, seeds, spices	3 (6.7%)
Vegetables	2(4.4%)
Plant foods of several of this groups	6 (13.3%)
Cofactors	11 (24.4%)
Exercise	9 (20%)
NSAIDs	2 (4.4%)
Symptoms	
Cutaneous symptoms	43 (95.6%)
Urticaria-angioedema	24 (53.3%)
Urticaria	8 (17.7%)
Exantema	11 (24.4%)
OAS	38 (84.4%)
Asthma	17.8%
Digestive symptoms	13.4%
Anaphylaxis	34 (75.6%)
Grade II	11(32.4%)
Grade III	8 (23.5%)
Grade IV	15 (44.1%)
Microarray	42/45
ISAC	14/45
ALEX	28/45
SPT Pru p3 mm,( mean (SD))	7.3 (SD 3)
SPT Granini^®^ (mm, mean (SD))	5.5 (SD 2.1)
LTP sensitization n (%)	45 (100%)
Profilin co-sensitization	5 (11.6%)
sIgE Pru p 3 kU/L, (mean (SD))	
ISAC	4.8 (9.8)
ALEX	6.45 (7.2)
FAQLQ-AF score	4.47

**Table 2 jcm-12-01823-t002:** SLIT and OIT outcomes.

PEACH ALK^®^ SLIT	Well Tolerated	Mild Reactions (OAS)	Moderate Reactions	Severe Reactions
N = 45	37 (82.2%)	5 (11.1%)	3 (6%)	0
GRANINI^®^ PEACH OIT				
N = 45	39 (86.6%)	5 (11.1%)	1 (2%)	0
FPT				
N = 45	39 (86.6%)	3 (6%)	2 (4%)	1 (2%)
		**Almonds**	**Almonds**	**Lentils**
		**Apple with peel**	**Apple with peel**	
		**Peanut**		

SLIT: sublingual immunotherapy; OIT: oral immunotherapy; OAS: oral allergy syndrome.

**Table 3 jcm-12-01823-t003:** Complementary data outcomes.

SPT	Baseline	Visit 4	Sig
Pru p3, mean (SD)	7.3 (3)	6.9 (2.2)	NS *
Granini^®^, mean (SD)	5.5 (2.1)	5.1 ( 2.5)	NS *
FAQLQ-AF score	4.47	2.7	*p* < 0.01 *
**SPT**	**Negative FPT**	**Positive FPT**	
SPT Pru p3	7.4 (2.9)	8.5 (3.3)	NS **
SPT Granini^®^	5.4 (1.9)	6.8 (2.6)	NS **
sIgE Pru p3	7.6 (11.7)	4.6 (4.9)	NS **

SPT: skin prick test; sIgE: specific IgE; FAQLQ-AF; food allergy quality of life questionnaire-adult form; FPT: final provocation test. * Wilcoxon test. ** Mann Whitney test.

## Data Availability

Data is unavailable due to privacy or ethical restrictions.

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
