# Peer review of "Combined LTP Sublingual and Oral Immunotherapy in LTP Syndrome: Efficacy and Safety"

_jcm, 2023, doi:10.3390/jcm12051823_

Round 1

Reviewer 1 Report

Dear Editor

many thankf for askinng me to review this original article aimed to evaluate the efficacy and safety of a new protocol starting 99 with SLIT-peach and following with OIT with commercial peach juice in patients with 100 LTP syndrome, and to assess the changes in the quality of life throughout the study. This study is a very well conducted and written study.

I have just an observation. In the Introduction section, I think that authors should briefly explain the mechanisms of immunotherapy in relationship to different routes of administration and cytokines network. Please, refer to "Cuppari C, et al. Allergen immunotherapy, routes of administration and cytokine networks: an update. Immunotherapy. 2014;6(6):775-786"

Author Response

Point 1: I have just an observation. In the Introduction section, I think that authors should briefly explain the mechanisms of immunotherapy in relationship to different routes of administration and cytokines network. Please, refer to "Cuppari C, et al. Allergen immunotherapy, routes of administration and cytokine networks: an update. Immunotherapy. 2014;6(6):775-786"

Response 1: after reviewing the recommended article and other referenced articles , I have added these sentences to the introduction, clarifying the mechanisms of immunotherapy.

“The mechanisms by which immunotherapy induces tolerance are unclear. OIT may alter the T cell responses by deletion of pro-allergic T cell responses, by the immune deviation from a Th2 to Th1, or by the induction of concurrent immune-regulating T cells. Dendritic cells may also play a role in OIT outcomes.

On the other hand,  current data support that antigens delivered by SLIT are taken up by a myeloid dendritic cell population in the oral mucosa, oral Langerhans cells. This promotes T-cell production of tolerogenic cytokines including IL-10 and TGF-β [26]. FOXP3- positive T regulatory cells are induced and Th2 cytokines including IL-4 are downregulated increasing Th1 citokines (INF-ϒ and IL-12).

Cuppari C, et al. Allergen immunotherapy, routes of administration and cytokine networks: an update. Immunotherapy. 2014;6(6):775-786"

Schworer SA,  Kim EH. Sublingual immunotherapy for food allergy and its future directions. Immunotherapy . 2020; 12: 921–931

Kulis MD,  Patil SU, Wambre E , Vickery BP. J Immune Mechanisms of Oral Immunotherapy. Allergy Clin Immunol. 2018 February ; 141: 491–498

Reviewer 2 Report

The manuscript by Martin Iglesias et al describes a combined LTP sublingual and oral immunotherapy method that proves to be adequate in the desensitization to peach LTP (Pru p 3) and cross- reactive proteins. Although the study is significant in the field and contributes to improving the diet and daily life of LTP-allergic patients and is described in detail, the way it is currently presented understates its importance.

Here are my suggestions to improve the readability of the manuscript and highlights its importance and merit of the researchers and physicians.

1/ I strongly suggests that the manuscript is revised by a native English speaker for English language and style.

2/ Please, include graphs and figures that summarize the methods and results obtained. A narrative text describing the findings is OK, but figures are more visual and eye-catchy. In this regard, the boxes containing the information of the visits should be grouped in a Figure.

3/ Consider making the introduction shorter since it is now quite long. The information contained is relevant, but maybe the style can be revised.

4/ Please, include an abbreviations list to clarify terms such as FAQLQ-AF.

5/ Although LTP allergy is mostly restricted to the Mediterranean area, there is a clear bias in the literature references from Spanish authors. In addition, are there more recent studies on the topic? Only two references are from 2022 and one from 2021. There are two recent EAACI reviews on LTPs that should also be included:

Skypala IJ, Asero R, Barber D, et al. Nonspecific lipidtransfer proteins: Allergen structure and function, crossreactivity, sensitization, and epidemiology. Clin Transl Allergy. 2021;113.

Skypala IJ, Bartra J, Ebo DG, et al. The diagnosis and management of allergic reactions in patients sensitized to non-specific lipid transfer proteins. Allergy. 2021;76:2433–2446.

7/ More specifically: What is the reason to not include placebo controls?

Round 2

Reviewer 2 Report

The manuscript quality has improved just minimally.

There are still a lot of typos that could have been revised before re-submission. English editing is mandatory before further re-submission.

Figure 1 and 2 are redundant. Being Figure 2 more visual, I would add the information about the paucity of SLIT & OIT contained in Figure 1 to Figure 2 and merge them.

Figure 3 does not really summarize the results. Where are the statistical comparisons? Please, provide a figure with graphs and statistical significance that supports the conclusions of your study.

Author Response

RESPONSE TO REVIEWER 2. ROUND 2

Point 1. The manuscript quality has improved just minimally.

There are still a lot of typos that could have been revised before re-submission. English editing is mandatory before further re-submission.

We send the revised text and also attach the English Editing Certificate

Point 2. Figure 1 and 2 are redundant. Being Figure 2 more visual, I would add the information about the paucity of SLIT & OIT contained in Figure 1 to Figure 2 and merge them.

I attach a Figure 1 with the general chronology of the study I hope it is eye-catchy, and little figures (2, 3 y 4) with the induction regimens of SLIT and peach juice and the final provocation test. I

Point 3. Figure 3 does not really summarize the results. Where are the statistical comparisons? Please, provide a figure with graphs and statistical significance that supports the conclusions of your study.

I am sending two new tables with the summary of the results supported by the statistical study
